# Determination of the Frequency of Migraine Attacks in Pregnant Women and the Ways They Cope with Headaches: A Cross-Sectional Study

**DOI:** 10.3390/healthcare11142070

**Published:** 2023-07-20

**Authors:** Guzin Kardes, Aytul Hadimli, Ahmet Mete Ergenoglu

**Affiliations:** 1Faculty of Health Science, Ege University, Izmir 35575, Turkey; guzin.kardes@ege.edu.tr; 2Faculty of Medicine, Ege University, Izmir 35100, Turkey; ahmet.mete.ergenoglu@ege.edu.tr

**Keywords:** pregnancy, migraine, headache, coping

## Abstract

One out of every five women of reproductive age suffers from migraine. Although headaches subside in most women during pregnancy, attacks continue and even worsen in some women. Pregnant women try to relieve pain with medication or non-pharmacological treatment methods. This descriptive and cross-sectional study was conducted to determine the incidence of migraine attacks in pregnant women diagnosed with migraine and the ways they cope with headaches. The study included 191 pregnant women who were diagnosed with migraine in the pre-pregnancy period. McNemar analysis was performed to test the relationship between descriptive statistical methods and categorical variables when the data were analyzed. The mean gestational age of the participants was 28.31 ± 8.64 weeks, and their mean age at the onset of migraine was 20.74 ± 5.63 years. The comparison of the duration, frequency, and severity of headaches suffered before and during pregnancy demonstrated that there were statistical differences between them (*p* < 0.05). The frequency of using methods such as taking painkillers, resting in a dark room, and having cold application and massage to relieve headaches before pregnancy decreased statistically significantly during pregnancy (*p* < 0.05). As a result, the frequency and severity of migraines decrease during pregnancy. The tendency to resort to pharmacological or non-pharmacological methods used to relieve headaches decreases during pregnancy. Although migraine has many adverse effects on pregnancy, pregnant women do not demand satisfactory information from health professionals about migraine headaches during pregnancy.

## 1. Introduction

Headache disorders are among the most common and disabling conditions worldwide [1,2]. Migraine, on the other hand, ranks second among the causes of disability and first among women under the age of 50 [3]. According to the comparison of white, black, and Hispanic people in terms of the prevalence of migraine or severe headache, the highest prevalence was among Native Americans and Alaska Natives (18.4%), and the lowest prevalence was among Asians (11.3%). The prevalence of migraine in the United States is 9.7% in men and 20.7% in women [4]. According to the results of studies conducted in Turkey, the prevalence of migraine is approximately 15% [5,6]. Although headache diseases do not lead to mortality, the pain experienced by people deteriorates their quality of life, reduces the workforce, and imposes a serious economic burden on society. During a headache attack, some people are unable to do their jobs and stop working, while others continue to work, but their productivity decreases by more than 50% [7,8,9]. Migraine headaches occur in attacks, and these attacks can last between 4 and 72 h in adults. The pain is often localized to one side of the head, but may also occur in the whole head. The pain that radiates from the nape or around the eyes is usually throbbing and restricts or prevents sufferers from doing their activities of daily living. Sufferers also experience nausea and vomiting, and discomfort from light and smell [10,11].

While the prevalence of migraine does not vary between the sexes in childhood, it increases sharply in women as adolescence begins. Migraine increases more prominently in women during childbearing, which is probably associated with changes in the levels of estrogen and progesterone hormones [12,13]. However, the specific mechanism is not entirely clear. It is stated that there are changes in headaches during menarche, menstruation, pregnancy, menopause, and oral contraceptive use, because all these conditions lead to a change in the estrogen level [14]. The increase in estrogen levels during pregnancy is reported as a factor that reduces headaches or causes them to disappear [12]. It is stated that in the first trimester of pregnancy, sudden hormonal changes increase headaches, and that in the second trimester, headaches decrease in 70–80% of women due to the increase in the estrogen level [12,13,15]. Especially in the second and third trimesters, many women with migraines have fewer headaches due to the stabilization of estrogen levels, increase in endorphin levels, and relaxation in the muscles; however, 4–8% of pregnant women with migraines may have worse headaches, and 10% of them have the first attack during pregnancy [13,15].

Migraine during pregnancy is associated with adverse outcomes in the mother and newborn, including pregnancy-induced hypertensive disorders, low birth weight, and preterm birth [16,17,18,19,20]. As was reported in a study conducted in Denmark, low birth weight, preterm birth, and cesarean delivery rates were higher in pregnant women suffering migraine, and in babies born to women with migraine, the duration of hospitalization was longer and respiratory problems were more common [21]. Therefore, controlling migraine attacks during pregnancy is of great importance.

The effects of most of the drugs used in the treatment of migraine on fetal development and pregnancy outcomes have not been adequately studied due to ethical concerns. Non-pharmacological techniques and treatments are the first-line treatment for women with mild migraines during pregnancy. In order to reduce migraine-related concerns during pregnancy, the use of non-pharmacological methods should be maximized. To do so, pregnant women should be encouraged to receive counseling services and to make lifestyle changes such as improving sleep duration and quality, having regular nutrition, consuming water or other liquids adequately, doing physical activity, and applying relaxation techniques. Here, the aim is to start these techniques in the pre-pregnancy period and to use them during pregnancy as early as possible. When non-pharmacological treatment is insufficient, it is stated that the use of behavioral techniques in combination with pharmacological treatments would be more effective [11,19,22,23]. Among the safe drugs that can be used in the acute treatment of migraine during pregnancy are acetaminophen (paracetamol), metoclopramide, and diphenhydramine (antiemetic and antihistaminic). Among non-steroidal anti-inflammatory drugs, the most reliable one is ibuprofen. In cases where analgesics do not work, triptanes are recommended [23,24,25].

Considering the high prevalence of migraine in the reproductive years and the negative effects of migraine on pregnancy, evaluation of the frequency of attacks in pregnant women with migraine gains importance. In the present study, we investigated the frequency of migraine attacks in pregnant women diagnosed with migraine and their ways of coping with headache.

## 2. Materials and Methods

### 2.1. Design and Setting

The study was carried out as a descriptive and cross-sectional study in order to determine the incidence of migraine attacks in pregnant women and the ways they used to cope with migraine.

### 2.2. Study Population and Sample

The population of the study carried out between October 2019 and September 2021 included pregnant women who presented to the Obstetrics Outpatient Clinic of the Ege University Medical Faculty Hospital, Department of Obstetrics and Gynecology. The number of pregnant women presenting to the obstetrics outpatient clinic of Ege University Medical Faculty Hospital in one year is approximately 7500. The sample size of the study was calculated as 190 in the Statcalc (EpiInfo Version 6) program, and 191 pregnant women were reached.

The inclusion criteria taken into account in determining the pregnant women to be included in the study sample were as follows:Having been diagnosed with migraine before pregnancy;Being in the second or trimester of pregnancy;Volunteering to participate in the study.

All the pregnant women who presented to the obstetrics clinic and were in their 2nd and 3rd trimesters were asked whether they had been diagnosed with migraine before pregnancy. Of them, those who answered “yes” were informed about the study, and those who volunteered to participate were included in the study. Migraine diagnosis and migraine type were evaluated in accordance with the headache criteria of the International Headache Society (IHS). Those who did not meet the migraine criteria were excluded from the study.

### 2.3. Ethical Issues

The present study was conducted in accordance with the principles of the Declaration of Helsinki. Before the study was conducted, ethical approval was obtained from the Clinical Research Ethics Committee of the Ege University Faculty of Medicine (Protocol Number: 19-9.1T/22). Permission to collect data from the participating pregnant women was obtained from the Faculty of Medicine, Department of Obstetrics and Gynecology. Pregnant women participating in the study were informed about the purpose and scope of the study, and their written informed consent was obtained.

### 2.4. Data Acquisition

The Sociodemographic and Obstetric Characteristics Questionnaire and the Migraine Headache Characteristics Determination Form were used to collect the study data. The data collection tools were administered to pregnant women diagnosed with migraine who met the inclusion criteria between October 2019 and September 2021. During the data collection process, the face-to-face interview technique was used.

According to the Migraine Diagnostic Criteria given in the International Classification of Headache Disorders 3rd edition, pregnant women were asked about the type of migraine they had before using the “Migraine Headache Characteristics Determination Form”, and the presence and type of migraine were confirmed in accordance with the International Classification of Headache Disorders 3rd edition, which is the latest classification of the International Headache Society [15].

Sociodemographic and obstetric characteristics questionnaire: The form developed by the authors in line with the literature [4,26,27,28,29] includes 9 items questioning the sociodemographic and obstetric characteristics of the participants.

Migraine headache characteristics determination form: This form developed by the authors in line with the literature in order to investigate the characteristics of headaches suffered by the participants before and during pregnancy [6,29,30,31,32] includes 23 questions (age at the diagnosis of migraine, family history of migraine, frequency, duration, severity of headache before and during pregnancy, onset of headache, what is done to relieve headache, amount of analgesics taken, factors triggering headache, receiving information from healthcare professionals about how to cope with migraine pain), whether the person regularly visits the physician for migraine treatment, etc.).

### 2.5. Statistical Analysis

The SPSS 26 (Statistical Package for Social Sciences) program was used for the statistical analysis of the data. McNemar analysis was performed to test the relationship between descriptive statistical methods (minimum and maximum values, arithmetic mean, standard deviation) and categorical variables in the analysis of the data.

## 3. Results

The mean age of the participating pregnant women was 29.72 ± 5.51 (min: 18; max: 42) years. Of them, all were married; 30.9% were high school graduates; and 74.3% were not working in a paid job. Their mean gestation age was 28.31 ± 8.64 (min: 14; max: 41) weeks. The mean number of pregnancies they had was 2.46 ± 1.43 (min: 1; max: 9). The mean number of births they gave was 1.01 ± 1.01 (min: 0; max: 4). The mean number of living children they had was 0.99 ± 0.96 (min: 0; max: 4). Of the participating pregnant women, 40.3% were in the second trimester and 59.7% were in the third trimester of their pregnancies. The most common diseases in pregnant women with migraine are gestational diabetes, preeclampsia, and abortion (Table 1).

The mean age of the participating pregnant women when they first had migraines was 20.74 ± 5.63 (min: 8; max: 37) years. Of them, 82.2% were diagnosed with migraine without aura; 56.5% stated that the headache started slowly; 75.4% stated that the pain was continuous; and 53.4% stated that they had at least one family member who had migraine. The location of the headache was unilateral in 84.8% of the pregnant women (Table 2).

While the trigger of migraine headaches was stress for 91.6% of pregnant women, it was noise and sound for 77.5% of them, insomnia for 77.0% of them, and bright light for 76.4% of them (Table 3).

Of the participating pregnant women, 45.0% were anxious about migraine headaches during pregnancy, 66.0% stated that their headaches decreased during pregnancy, and the rates of the participants who did not have regular doctor checks regarding migraine treatment before and during pregnancy were 86.4% and 98.4%, respectively (Table 4).

A statistically significant difference was determined between the pre-pregnancy period and the pregnancy period in terms of the frequency, duration, and severity of headaches (*p* < 0.05) (Table 5).

A statistically significant difference was determined between the pre-pregnancy period and the pregnancy period in terms of using pain relief methods such as analgesics, resting in a dark room, sleeping, applying cold to the head, and having a massage (head and neck) (*p* < 0.05) (Table 6).

Of the other methods mentioned in Table 7, the ones used to relieve pain before and during pregnancy were applying pressure to the forehead by tying a scarf tight around the head and rubbing peppermint oil, cologne, and migraine stones on the forehead and temples (Table 7).

## 4. Discussion

Migraine usually begins during childhood or adolescence. From the reproductive age onwards, its prevalence in women increases compared to men [30,32,33,34,35]. In the present study, the mean age of the participants at the onset of migraine was similar to that in the literature. Of the pregnant women who participated in our study, 53.4% stated that they had a family history of migraine in at least one of their family members. Migraine is a familial disease [36,37,38]. As reported in several studies, the prevalence of a family history of migraine in patients with migraine ranges from 33.1 to 78.3%, which is consistent with our findings [29,39,40,41,42,43,44].

Evidence indicating that there is a relationship between migraine during pregnancy, preterm birth risk, and vascular diseases such as gestational hypertension and preeclampsia is available [18,45,46,47,48]. In our study, we determined that 20.4% of the pregnant women had at least one disease in their current pregnancy. Of them, eight were diagnosed with gestational diabetes, five with imminent abortion, five with preeclampsia, five with urinary tract infection, three with preterm birth threat, and three with placenta previa.

Of the pregnant women who participated in our study, 82.2% were diagnosed with migraine without aura. Since the criteria for migraine without aura include unilateral headaches, the majority of the participants in the study had unilateral headaches. In studies, the most common complaints were migraine without aura [26,49,50,51] and unilateral headache [29,52,53]. One of the migraine criteria of the International Headache Society is throbbing pain [1]. As in the present study, in many studies, women defined the character of the pain as throbbing [27,39,52,54].

In many cross-sectional and clinical studies, stress, changes in the menstrual cycle, weather changes, sleep disorders, alcohol, and some foods are shown as triggering factors for migraine attacks, and it is recommended that sufferers stay away from these triggers [55]. In our study, as in the literature, according to the statements of the participating pregnant women, stress, insomnia, noise/sound, bright light, and hunger were the factors triggering migraine headaches the most.

If a pregnant woman has a headache complaint, the priority should be the separation of the primary causes (such as migraines, tension headaches, and cluster headaches) from serious secondary causes (such as preeclampsia and cerebral vein thrombosis) [56]. Pregnancy is considered a positive period for women with migraine, as it eliminates the fluctuation of hormone levels, which is a leading triggering factor for migraine attacks [57,58].

Both prospective studies conducted during pregnancy and retrospective studies performed at the end of the third trimester or shortly after delivery in women with migraine showed that in 30–80% of the patients, the frequency and severity of attacks of migraine headaches decreased [12,41,59,60,61]. Such a decrease is more common, especially in those with migraine without aura [59,62,63]. A statistically significant difference was determined between the pre-pregnancy and pregnancy periods in terms of the frequency and severity of headaches suffered by the participants. Our result that headaches subside in the second and third trimesters of pregnancy is consistent with the results of other studies in the literature [41,54,64].

The review of the results obtained from the controls related to migraine of the participating pregnant women demonstrated that of them, 13.6% received information about how to cope with migraine pain during pregnancy, and 13.6% and 1.6% had regular controls for migraine treatment before and during pregnancy, respectively. The rate of receiving information from healthcare professionals about migraine before and during pregnancy, and the rate of having regular medical check-ups for migraine treatment are low, both in other studies and in the present study [46,52,65,66]. The low rates obtained in the present study can be explained by the fact that headaches decrease during pregnancy and that the study was carried out during the COVID-19 pandemic. The comparison of the pre-pregnancy and pregnancy periods revealed a significant difference in the use and amount of analgesics. This decrease is thought to result from the fact that drug use is avoided during pregnancy and that the frequency of headaches decreases. In migraine management, it is recommended that lifestyle changes such as sleep hygiene, exercise, and stress management be made and that factors that trigger attacks be avoided [67,68]. Evidence accumulated in recent years has revealed that the regular use of non-pharmacological treatment methods such as biofeedback and muscle relaxation can provide a 45–60% reduction in the frequency and severity of migraine [25,69,70]. In several studies, the effectiveness of acupuncture in the treatment of migraine has been demonstrated. In the Cochrane Collaboration review, which included 22 randomized controlled trials, it was stated that acupuncture was at least as effective as prophylactic drug therapy and that it had fewer side effects [71]. In studies in which the effect of using acupuncture in migraine prophylaxis during pregnancy and in reducing the intensity and frequency of migraine attacks, nausea, and vomiting in the first trimester was investigated, it was concluded that acupuncture was safe and effective [71,72]. In the present study, only two pregnant women stated that they underwent acupuncture in the pre-pregnancy period; however, they did not undergo this method during pregnancy. There was a statistically significant difference between the pre-pregnancy and pregnancy periods in terms of using the most common coping methods, such as resting in a dark room, sleeping, and having cold applications and massages. The methods used less frequently by the pregnant women were applying pressure to the forehead by tying a scarf tight around the head, rubbing peppermint oil and migraine/menthol stones on the forehead and temples, and consuming herbal tea. Although there is no evidence on whether they are safe to use in pregnancy, of the herbal products, chamomile and peppermint were used more frequently [73,74,75]. However, our search for studies in which the use of peppermint and other aromatic oils during pregnancy is safe revealed a gap in the literature. In line with these findings, we can say that more studies investigating the effectiveness of non-pharmacological methods in the management of migraine symptoms during pregnancy should be conducted.

## 5. Conclusions

Although migraine has several adverse effects on pregnancy, both pregnant women and health professionals (gynecologists, obstetricians, and midwives) mostly focus on fetal health or other health problems of the mother. Pregnant women do not demand satisfactory information from health professionals about migraine headaches during pregnancy. Therefore, migraine should also be included in the risk assessment during pregnancy. Health professionals should inform pregnant women about migraines and encourage them to obtain information.

In addition, pregnant women should be informed about healthy lifestyle changes (regular diet, adequate sleep, stress management, exercise, and smoking cessation) and evidence-based pain relief methods (acupuncture and relaxation methods), which can reduce the frequency of migraine attacks. It may be recommended to conduct more studies on the effectiveness of the methods (migraine stones, peppermint oil, and herbal tea consumption) that pregnant women use during pain.

### Limitations of the Study

Because the present study was cross-sectional, the data obtained are applicable only to the period when the study was conducted and may vary over time. The study was carried out with women in the second and third trimesters of their pregnancy, and we compared these periods with the pre-pregnancy period. The questions about the pre-pregnancy period were retrospective and included evaluation based on the memory factor, which constituted another limitation.

## Figures and Tables

**Table 1 healthcare-11-02070-t001:** Obstetric characteristics of the participating pregnant women.

Obstetric Characteristics		Mean ± SD/n (%)
Mean gestational age (weeks)		28.31 ± 8.64
Mean number of pregnancies		2.46 ± 1.43
Mean number of births		1.01 ± 1.01
Mean number of living children		0.99 ± 0.96
Pregnancy trimesters	Second trimester	77 (40.3)
	Third trimester	114 (59.7)
Having sickness during pregnancy	Yes	39 (20.4)
	No	152 (79.6)
Name of the sickness *	Imminent abortion	5 (2.6)
	Premature birth risk	3 (1.6)
	Hyperemesis gravidarum	2 (1.0)
	Placenta previa	3 (1.6)
	RH incompatibility	2 (1.0)
	Preeclampsia	5 (2.6)
	Gestational diabetes	8 (4.2)
	Urinary tract infection	5 (2.6)
	Hypothyroidism	1 (0.5)
	Hyperthyroidism	2 (1.0)
	Cholestasis	1 (0.5)
	Toxoplasmosis	1 (0.5)
	Rubella	1 (0.5)
	COVID-19	2 (1.0)

* Some of the participating pregnant woman had more than one disease during pregnancy.

**Table 2 healthcare-11-02070-t002:** Pre-pregnancy migraine headache-related characteristics of the women participating in the study.

Migraine Headache-Related Characteristics		n (%)
Mean age at the onset of migraine		20.74 ± 5.63
Migraine type	Migraine without aura	157 (82.2)
	Migraine with aura	34 (17.8)
How the migraine begins	Suddenly	62 (32.5)
	Gradually	108 (56.5)
	Erratically	21 (11.0)
Duration of the headache	Continuous	144 (75.4)
	Varying	47 (24.6)
Location of the headache	Unilateral	162 (84.8)
	Bilateral	13 (6.8)
	Others	16 (8.4)
Time of aura	Before the pain	11 (32.4)
	During the pain	22 (64.7)
	After the pain	1 (2.9)
Family history of migraine	Yes	102 (53.4)
	No	89 (46.6)

**Table 3 healthcare-11-02070-t003:** Factors triggering pre-pregnancy migraine headaches in the women participating in the study.

Factors Triggering Migraine Headaches		n (%)
Stress	Yes	175 (91.6)
	No	16 (8.4)
Noise/sound	Yes	148 (77.5)
	No	43 (22.5)
Insomnia	Yes	147 (77.0)
	No	44 (23.0)
Bright light	Yes	146 (76.4)
	No	45 (23.6)
Hunger	Yes	136 (71.2)
	No	55 (28.8)
Humidity/heat/cold air	Yes	118 (61.8)
	No	73 (38.2)
Menstruation	Yes	71 (37.2)
	No	120 (62.8)
Foods and beverages	Yes	49 (25.7)
	No	142 (74.3)

**Table 4 healthcare-11-02070-t004:** Pregnancy and migraine-related experiences.

Migraine-Related Experiences		n (%)
Concern about experiencing migraine headaches during pregnancy	Yes	86 (45.0)
	No	105 (55.0)
Change in the severity of headaches during pregnancy	Decreased	126 (66.0)
	Remained the same	28 (14.7)
	Increased	37 (19.3)
Receiving information from healthcare professionals about how to cope with migraine pain during pregnancy	Yes	26 (13.6)
	No	165 (86.4)
Having regular doctor check-ups due to the diagnosis of migraine in the pre-pregnancy period	Yes	26 (13.6)
	No	165 (86.4)
Having regular doctor check-ups due to the diagnosis of migraine during pregnancy	Yes	3 (1.6)
	No	188 (98.4)

**Table 5 healthcare-11-02070-t005:** Comparison of headache-related characteristics of the participating pregnant women before and during pregnancy.

Frequency of Headaches	More Than Once a Month during Pregnancy	Once a Month and Less Often Than Once a Month during Pregnancy	Total (n)	
More than once a month before pregnancy	53	63	116	X^2^ = 14.606 * *p* = 0.000 **
Once a month and less often than once a month before pregnancy	14	61	75	
Total	67	124	191	
Severity of Headaches	No pain and mild pain during pregnancy	Severe and very severe pain during pregnancy	Total (n)	X^2^ = 14.194 * *p* = 0.000 **
Mild pain before pregnancy	41	12	53	
Severe and very severe pain before pregnancy	65	73	138	
Total	106	85	191	

* X^2^_McNemar_ ** *p* < 0.05.

**Table 6 healthcare-11-02070-t006:** Comparison of the methods used to relieve headaches before and during pregnancy.

Using Analgesics before Pregnancy	Using Analgesics during Pregnancy			
	Yes	No	Total (n)	X^2^ = 10.246 * *p* = 0.000 **
Yes	48	106	154	
No	2	35	37	
Total	50	141	191	
Resting in a Dark Room Before Pregnancy	Resting in a Dark Room During Pregnancy			X^2^ = 62.161 * *p* = 0.000 **
	Yes	No	Total (n)	
Yes	126	32	158	
No	3	30	33	
Total	129	62	191	
Sleeping Before Pregnancy	Sleeping During Pregnancy			
	Yes	No	Total (n)	X^2^ = 80.887 * *p* = 0.000 **
Yes	109	29	138	
No	4	49	53	
Total	113	78	191	
Cold Application Before Pregnancy	Cold Application During Pregnancy			
	Yes	No	Total (n)	X^2^ = 1.008 * *p* = 0.027 **
Yes	40	16	56	
No	5	130	135	
Total	45	146	191	
Having Massage Before Pregnancy	Having Massage During Pregnancy			
	Yes	No	Total (n)	X^2^ = 80.708 * *p* = 0.000 **
Yes	81	33	114	
No	4	73	77	
Total	85	106	191	

* X^2^_McNemar_ ** *p* < 0.05.

**Table 7 healthcare-11-02070-t007:** Other methods used to relieve headaches before and during pregnancy.

	Before Pregnancy	During Pregnancy
Applying pressure to the forehead by tying a scarf tight around the head	9 (4.7)	5 (2.6)
Rubbing peppermint oil on the forehead and temples	9 (4.7)	4 (2.1)
Rubbing cologne on the forehead and temples	6 (3.0)	3 (1.5)
Rubbing migraine/menthol stones on the forehead and temples	5 (2.6)	5 (2.6)
Having herbal tea	5 (2.6)	1 (0.5)
Acupuncture	2 (1.0)	-
Consuming caffeine	1 (0.5)	2 (1.0)
Bloodletting (withdrawing blood from a person’s veins for therapeutic reasons)	1 (0.5)	-
Oxygen therapy	1 (0.5)	-

## Data Availability

Data sharing is not applicable.

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
