# Peer review of "Determination of the Frequency of Migraine Attacks in Pregnant Women and the Ways They Cope with Headaches: A Cross-Sectional Study"

_healthcare, 2023, doi:10.3390/healthcare11142070_

Round 1
Reviewer 1 Report
Row 101: How was the diagnosing of migraine? A lot of patients with tension type headache postulates they suffer from migraine! Did your patients fulfil the criteria of the Headache Classification Committee of the International Headache Society (IHS). For accepting this article, I find it mandatory that the migraine diagnosing has been performed in a fully professional way, either by a neurologist or by another specialist clearly following the IHS classification. Thus, please describe how!
Row 117-18: ‘Migraine Headache Characteristics Determination Form’, any reference? What is this?
Row 126: ‘Researchers’, do you mean ‘the Authors’?
Row 125-27: Specify the 23 questions concerning type of headache!
Row 209-11: An important statement!
Row 215-222: How do you present the quantitative differences. You only report subjective experiences. Are they included in the 23 questions?
Row 220-22: Severity of headache, where is it described?
Row 232-34: How do you measure consumption of analgesics?
Row 252-61: I do not see the point here, is it a question of placebo or what?
Row 263-67: It seems contradictory? I guess, it could be a consequence of unfamiliarity with the English language, a problem which I recognize for myself and is so well aware of!
Author Response
Dear Reviewer,
We would like thank you very much for your suggestions and contribution to the manuscript ID healthcare-2447106. Please see the attachment.
Kind regards

Reviewer 2 Report
In the version of the paper supplied to me I find the contents of Tables 5 and 6 very difficult to interpret. It is quite possible that the contents of some cells have got into the wrong rows, and in particular this may apply to the top rows of the tables, but the result is that it becomes impossible to determine on what denominator some percentages have been calculated and in Table 5 it would be possible to conclude that the majority of the women did not have migraine attacks before pregnancy. I think this matter has to be clarified.
I note there is no mention of whether the women were taking continuous migraine prevention drug therapy before or during pregnancy, or whether they were using specific anti-migraine agents such as triptans before or in pregnancy, and this might be quite relevant to migraine frequency and severity in the two sets of circumstances.
The use of English in the paper goes astray in a few places, for example at the end the text when the authors thank their patients for allowing them to interview the authors, rather than vice versa.
Author Response

(The authors gave the same response as above.)

Round 2
Reviewer 2 Report
I think the versions of your Tables 5 and 6 are now very much more easily followed than in your previous version, but in your left hand columns of these tables you show (n) without any accompanying numerical value, with the relevant value appearing in the extreme right-hand column. If in the left column you showed, instead of `n' only (`n = the number in the right column) you could get rid of the right-hand column and it would be even easier for the reader to follow.
In some of your earlier Tables it is difficult to be sure whether you are referring to the characters of the patient's migraine before or during pregnancy, except that in one place you mentioned that menstruation was a provoking factor for attacks.
On the possible limitation section of your paper you did not go into the matter of whether there might be bias in selection of patients studied or whether all possible candidates for study were invited to participate and then (stated) exclusion criteria applied, and whether those who did not wish to participate were invited to provide reasons.
How did you categorise women who sometimes get migtaine with aura and at other times migraine without aura? Such migraine sufferers are not rare, but I admit many oublications ignore the issue.
It seems a pity that patients were not questioned about taking migraine specific pharmacological agents such as triptans and continuing migraine prophylactic therapy. I think prospective readers will query the absence of this information I would have been very interesting to know if such therapy had been used before pregnancy and been ceased during it.
In at least two places in the paper I note statements that may not convey their intended meaning to readers. These are:
1. lines 207-208 seem to state that because the headache in migraine without aura tends to be unilateral most of your patients had unilateral headache. I imagine that you must have asked them about the situation where their headache occurred and that you did not decide from the type of headache where the headache was situated.
2. At the conclusion of your text you thank the patients who cooperated and stated that the information they provided was `reliable'. What evidence did you have that the information was reliable? Perhaps you meant `relevant'.
Overall,the English in the paper Is not difficult to follow, but there are a couple of statements made which may be interpreted in a way that the authors may not intend. I will point them out to the authors in my comments directed to them.
Author Response
Dear Reviewer,
We would like thank you very much for your suggestions and contribution to the manuscript ID healthcare-2447106. Please see the attachment.
Thank you
